# Impact of Antimicrobial-Resistant Bacterial Pneumonia on In-Hospital Mortality and Length of Hospital Stay: A Retrospective Cohort Study in Spain

**DOI:** 10.3390/antibiotics14101006

**Published:** 2025-10-10

**Authors:** Iván Oterino-Moreira, Montserrat Pérez-Encinas, Francisco J. Candel-González, Susana Lorenzo-Martínez

**Affiliations:** 1Microbiology and Parasitology Department, Faculty of Pharmacy, Complutense University of Madrid, 28040 Madrid, Spain; 2Pharmacy Department, Móstoles University Hospital, 28935 Móstoles, Spain; 3Pharmacy Department, Alcorcón Foundation University Hospital, 28922 Alcorcón, Spain; 4Microbiology Department, San Carlos Clinical Hospital, 28040 Madrid, Spain; 5Quality and Patient Management Department, Alcorcón Foundation University Hospital, 28922 Alcorcón, Spain

**Keywords:** antimicrobial resistance, international classification of diseases, bacterial pneumonia, administrative database

## Abstract

Objectives: Antimicrobial resistance is a major global health threat. This study aimed to assess the impact of antimicrobial-resistant bacterial pneumonia on in-hospital mortality and length of hospital stay in Spain using a large, nationally representative cohort. Methods: A retrospective cohort study that used data from Spain’s Registry of Specialized Health Care Activity (RAE-CMBD) between 2017 and 2022. Hospitalized adults with bacterial pneumonia were included. Hospitalization episodes with bacterial antimicrobial resistance, defined according to ICD-10-CM codes for antimicrobial resistance (Z16.1, Z16.2), were analyzed versus hospitalization episodes without these codes. Multivariate logistic regression models, adjusted for potential confounders (e.g., age, comorbidity, intensive care unit admission) and sensitivity analyses (Poisson regression and propensity score matching test), were performed. Results: Of the 116,901 eligible hospitalizations, 6017 (5.15%) involved antimicrobial-resistant bacteria. Patients with antimicrobial-resistant bacterial pneumonia were older (median 75 vs. 72 years), had greater comorbidity (Elixhauser–van Walraven index: 8 vs. 5), and were more frequently admitted to the intensive care unit (22% vs. 14%). Crude in-hospital mortality was higher in the antimicrobial resistance group (18.46% vs. 10.05%, *p* < 0.0001), with an adjusted odds ratio of 1.47 (95% confidence interval, 1.36–1.58), *p* < 0.0001. Length of hospital stay was prolonged in antimicrobial resistance patients (median 14 vs. 8 days; adjusted incident rate ratio of 1.46; 95% confidence interval of 1.41 to 1.50). The most prevalent antimicrobial resistant pathogens were *Staphylococcus aureus* and Gram-negative bacilli (*Pseudomonas aeruginosa*, *Klebsiella pneumoniae*, and *Escherichia coli*). Conclusions: Antimicrobial resistance is associated with longer hospital stays and an up to 50% higher risk of mortality. Despite the implementation of control policies in place over the past decade, policymakers must strengthen AMR surveillance and ensure adequate resource allocation. Clinicians, in turn, must reinforce antimicrobial stewardship and incorporate rapid diagnostic tools to minimize the impact of antimicrobial resistance on patient outcomes.

## 1. Introduction

Antimicrobial resistance (AMR) is a slow-moving and silent pandemic that poses a major threat to global health and economic stability [1].

There are four major categories of antimicrobial resistance exhibiting microorganisms: multidrug-resistant (MDR), extensively drug-resistant (XDR), pan drug-resistant (PDR), and difficult-to-treat resistance (DTR). MDR microorganisms acquired non-susceptibility to at least one agent in three or more antimicrobial classes. XDR microorganisms are non-susceptible to at least one agent in all but two or fewer antimicrobial classes. Microorganisms with non-susceptibility to all agents in all antimicrobial classes are referred to as PDR. Finally, difficult-to-treat resistance (DTR) is characterized by non-susceptibility to all first-line antimicrobials, including beta-lactams and fluoroquinolones, thereby restricting limiting treatment options to more toxic and/or less effective antimicrobials [2].

The World Health Organization (WHO) estimated that bacterial AMR was directly responsible for 1.27 million deaths worldwide in 2019 and contributed to an additional 4.95 million deaths [3].

The Centers for Disease Control and Prevention (CDC) reported that each year more than 2.8 million AMR infections occur in the United States (U.S.), resulting in over 35,000 deaths [4]. In the European Union (EU), this corresponds to approximately 25,000 deaths per year [1]. In Spain, 4000 people die each year, four times more than the number of deaths caused by traffic accidents in 2021 [5].

Although poorly quantified, the global burden of AMR is likely concentrated in three major areas: (i) prolonged illness and higher mortality rates in patients with resistant infections; (ii) increased treatment costs for such infections; and (iii) the inability to perform medical procedures that depend on effective antimicrobial prophylaxis [6]. Barrasa-Villar et al. concluded that hospital mortality from all-cause hospital mortality was 1.7 times higher in infections caused by multidrug-resistant organisms compared to those caused by susceptible microorganisms [7].

Various policy frameworks have been established to address the growing challenge of AMR. “The Global action plan on antimicrobial resistance” to tackle the growing problem of antimicrobial medicines, endorsed by the World Health Assembly in May 2015, provided a strategic framework for developing national action plans, outlining key measures to be implemented within 5–10 years by multiple stakeholders to combat AMR [8].

Several countries have subsequently developed their own action plans to combat this “*silent pandemic*” such as “*A European One Health Action Plan against Antimicrobial Resistance*” (EU since 2011) or the “*National Action Plan for Combating Antibiotic-Resistant Bacteria*” (U.S. since 2014) [1,4].

Given the transmission of bacterial pathogens, the health of humans, domestic and wild animals, plants, and the wider environment (including ecosystems) are closely interconnected. This recognition led to the development of the “One Health” concept by the WHO, later adopted globally as an integrated and unifying approach that seeks to sustainably balance and optimize the health of people, animals, and ecosystems [9].

The *One Health* joint plan of action (2022–2026) of the WHO, the Agriculture Organization of the United Nations (FAO), the World Organization for Animal Health (WOAH), and the (United Nations Environment Programme UNEP) was established as a quadripartite cooperation. Its purpose is to provide coordinated support for the integration of the One Health approach worldwide. Among its priority action tracks, it specifically addresses the challenge of antimicrobial resistance (AMR) as a “*silent pandemic*” and emphasizes the integration of environmental considerations into *One Health* strategies [10].

The 2024 WHO Bacterial Priority Pathogens List (WHO BPPL) is an important tool in the global response to antimicrobial resistance. The updated list identifies 24 bacterial pathogens of greatest concern, with the aim of addressing the evolving challenges of AMR and guiding priorities in research, the development of new antimicrobials, and public health interventions [11].

The WHO BPPL stratified the most worrying bacterial pathogens worldwide into three priority groups [11].

*Critical priority group*—The highest threat to public health due to limited treatment options: *Enterobacterales* carbapenem-resistant, *Enterobacterales* third-generation cephalosporin-resistant, *Acinetobacter baumannii* carbapenem-resistant, and *Mycobacterium* tuberculosis rifampicin-resistant.*High priority group*—AMR pathogens that are significantly difficult to treat: *Salmonella typhi* fluoroquinolone-resistant, *Shigella* spp. fluoroquinolone-resistant, *Enterococcus faecium* vancomycin-resistant, *Pseudomonas aeruginosa* carbapenem-resistant, non-typhoidal *Salmonella* fluoroquinolone-resistant, *Neisseria gonorrhoeae* third-generation cephalosporin and/or fluoroquinolone-resistant, and *Staphylococcus aureus* methicillin-resistant (MRSA).*Medium priority group*—AMR pathogens that are associated with moderate difficulty for treatment): Group A Streptococci macrolide-resistant, *Streptococcus pneumoniae* macrolide-resistant, *Haemophilus influenzae* ampicillin-resistant, and Group B Streptococci penicillin-resistant.

Pneumonia is one of the leading causes of hospital admissions in the U.S. and remains a major cause of death [12]. In 2023, the CDC reported 41,210 deaths in the U.S. due to pneumonia, with a crude death rate of 12.3 per 100,000 population [13].

Some authors have estimated the mortality rate of community-acquired pneumonia in hospitalized patients to be approximately 15%. However, the emergence of resistant pathogens complicates management [14].

The WHO recognizes that although recent studies position AMR as one of the leading causes of death worldwide, the exact morbidity and mortality associated with AMR are very difficult to establish, and, in many settings, reliable estimates are unavailable. These knowledge gaps highlight the need to foster studies on AMR attributable mortality and morbidity [15]. The U.S. and EU action plan for combating AMR include among their goals the need for new research and evidence-based analysis and data to better understand the risks of AMR [1,4].

Since the causative pathogen is identified in fewer than 40% of pneumonia cases and is bacterial in less than 14% of isolates, most studies conducted in hospitalized patients with pneumonia include a limited sample of patients with bacterial isolates [7,16].

Large-scale epidemiological studies using real-world administrative data are a high-quality source of information on the true impact of AMR in a population. Administrative databases, such as Spain’s Registry of Specialized Health Care Activity (RAE-CMBD), provide a valuable resource for analyzing the impact of AMR on health [17].

This study, based on a large national cohort of patients with bacterial pneumonia and isolation of the responsible pathogen over six years (2017–2022), analyzed the real impact of AMR-related bacterial pneumonia on in-hospital mortality and LOHS in Spain. The primary objective was to determine the impact of AMR on in-hospital mortality in patients with bacterial pneumonia. Secondary objectives included assessing differences in hospital stay duration, identifying others risk factors associated with increased in-hospital mortality and prolonged hospitalization, and evaluating trends in the prevalence of AMR bacterial pneumonia requiring hospital admission in Spain.

## 2. Results

The total number of hospitalizations of patients over 18 years of age in Spain recorded in the national RAE-CMBD from 2017 to 2022, except hospitalizations for childbirth (1,857,052), was 22,331,410 [18]. Of these, 3,664,087 hospitalizations (16.40%) corresponded to bacterial infections.

We identified 186,800 (5.10%) episodes of hospitalization of patients with bacterial pneumonia diagnosis (excluding *Mycobacteriales*/*Spirochaetales* orders). After applying the inclusion and exclusion criteria, 116,901 hospitalizations were confirmed as eligible for this study.

The median age at hospital admission was 73 years (IQR: 60–83). The country of birth was Spain in 79.43% of cases (92,852). The hospitalizations with the code for AMR bacteria represented 5.15% (6017) of all bacterial pneumonia hospitalizations. Table 1 shows the prevalence of AMR across the years of study.

The demographic and clinical characteristics in hospitalized adults with registry of the AMR bacteria code are shown in Table 2. The AMR bacteria group were significantly older: median 75 years (IQR: 64–84) versus (vs.) 72 years (IQR: 69–83), respectively (*p* < 0.0001); and had a significantly higher comorbidity [median of EVCI 8 (IQR: 3–15) vs. 5 (IQR: 0–12), *p* < 0.0001]. ICU admission, mechanical ventilation, sepsis, ventilator-associated pneumonia (VAP), nosocomial condition, and extracorporeal membrane oxygenation (ECMO) were also significantly higher for the AMR bacteria group (Table 2).

According to the APR-DRG severity classification, 43.73% of AMR patients were categorized as “*extreme severity*” vs. 24.32% of non-AMR patients (*p* < 0.0001). The risk of mortality by APR-DRG was also significantly higher in the AMR group, with 30.16% ranked as “*extreme risk*” vs. 19.45% in the non-AMR population (*p* < 0.0001). Patients with AMR bacterial infection had more fungal pneumonia co-infection (Table 2).

Crude analysis showed that patients with AMR bacterial pneumonia had a median LOHS of 14 days (IQR: 8–24) vs. 8 days (5–13) in patients without AMR bacterial infection (*p* < 0.0001). The median of length of ICU stay was also significantly longer: 13 days (IQR: 5–26) versus 8 days (IQR: 3–17) (*p* < 0.00001). Hospitalized patients with AMR bacterial pneumonia had significantly higher in-hospital mortality: 18.46% vs. 10.05% (*p* < 0.0001) (Table 2).

The microbial etiology of pneumonia, according to the ICD-10-CM classification, is detailed in Table 3. *Streptococcus pneumoniae* was the most frequent microorganism, representing 60.15% of all bacterial pneumonias, followed by *Pseudomonas aeruginosa* (7.10%), *Legionella pneumophila* (6.40%), *Staphylococcus aureus* (5.29%), *Haemophilus influenzae* (4.82%), other Gram-negative aerobic bacteria (4.15%), *Klebsiella pneumoniae* (3.32%), and *Escherichia coli* (2.58%).

As shown in Table 3, the Gram-positive cocci group and Gram-negative bacilli group represent most AMR bacteria (57.20% and 41.78%, respectively), while the group “other bacteria” barely included AMR bacteria (1.01%). By microbial etiology, *Staphylococcus aureus* had the most AMR bacteria (43.28% of the isolates), followed by *Escherichia coli* (15.20%), *Klebsiella pneumoniae* (14.39%), and *Pseudomonas aeruginosa* (10.35%).

To estimate the risk factors associated with higher mortality, we calculated the odds ratio (OR) from the univariate logistic regression models (Figure 1). These risk factors were treated as potential confounders (covariates) in the multivariate logistic regression model posterior.

Adjusted analysis showed that pneumonia due to AMR bacteria increased in-hospital mortality: OR 1.47 (95% CI 1.36–1.58), *p* < 0.0001 (Figure 2). Older age and higher comorbidity [OR 4.11 (95% CI 3.80–4.44) EVCI quintile 5 vs. EVCI quintile 1 and OR 3.54 (95% CI 3.31–3.79) quintile 5 vs. quintile 1, respectively], were the factors that most influenced mortality, followed by, sepsis [OR 2.71 (95% CI 2.54–2.84)], fungal pneumonia co-infection [OR 2.69 (95% CI 2.38–3.02)], viral pneumonia co-infection [OR 1.96 (95% CI 1.84–2.08)], and VAP OR 2.04 [(95% CI 1.79–2.33)], all *p* < 0.0001.

In the adjusted model, pneumonia due to AMR bacteria was also associated with prolonged hospital stay. The probability of a LOHS longer than 8 days (median LOHS of the population) was OR 2.87 (95% CI 2.70–3.05), *p* < 0.0001. Other significant predictors of prolonged stay included VAP, fungal pneumonia co-infection, comorbidity, sepsis, viral pneumonia, and age (Figure 3).

In the subgroup analysis by adjusted model (Table 4), patients with pneumonia due to AMR Gram-positive cocci and AMR Gram-negative bacilli showed increased mortality, with OR of 1.62 (95% CI 1.48–1.78), *p* < 0.000, and OR 1.41 (95% CI 1.26–1.58), *p* < 0.0001, respectively.

### Sensitivity Analysis

Multivariate Poisson regression analysis showed an increased incidence rate ratio (IRR) of both mortality and LOHS in hospitalizations with pneumonia due to AMR bacteria compared to the non-AMR group: IRR 1.34 (95% CI 1.26–1.41), *p* < 0.0001, and IRR 1.46 (95% CI 1.41–1.50), *p* < 0.0001, respectively (Table 5).

Propensity score matching using logistic regression showed an increase in both in-hospital mortality and LOHS associated with AMR bacterial pneumonia compared to hospitalizations without the AMR bacterial pneumonia group (average treatment effect on the treated, ATET) and in the overall population (average treatment effect, ATE) (Table 6). The standardized mean differences before and after matching for ATET in relation to in-hospital mortality are shown in Appendix A.

## 3. Discussion

Pneumonia due to AMR bacteria increased in-hospital mortality: OR 1.47 (95% CI 1.36–1.58), *p* < 0.0001. To our knowledge, this study includes one of the largest cohorts to evaluate the association between AMR bacterial infection and in-hospital mortality in cases of bacterial pneumonia requiring hospitalization.

The 5.15% prevalence of AMR bacterial pneumonia in our cohort is consistent with that reported in Spain by Aliberti et al. (7.6%) [19]. Crude mortality was nearly twice as high in AMR cases (18.46% vs. 10.05%, *p* < 0.0001), in line with previous reports of bacterial pneumonia requiring hospitalization, which mortality rates of 10–15% [14,20,21,22].

Based on these data, adjusted analyses estimated that AMR infection increases the risk of mortality by 37% to 40% [23], which represents a mortality rate between 3.7% to 6.0% higher that of the reference group (without AMR), supported by a propensity score matching test (ATE 3.82% and ATET 4.01%) and a Poisson adjusted regression [IRR 1.34, (95% CI 1.26–1.41)]. Similar effects were observed internationally: Lakbar et al. reported OR 1.39 for ICU- mortality adjusted [24]; while Lambert et al. reported that the risk of death associated with antimicrobial resistance (additional risk of death to that of the infection) in patients with pneumonia admitted to the ICU was 1.2 (95% CI 1.1–1.4), with no *p*-value reported [25].

Wanda et al. also found an increase in ICU mortality due to AMR bacterial infection: OR 1.65 (95% CI 1.18–2.43), without making a distinction between the type of infection [26]; Hixon et al. calculated a median excess mortality difference of 5.8% (IQR: 2.6–4.0%) in cases of AMR Gram-negative bacterial pneumonia [27]; and Nelson et al. reported mortality attributable to health care-associated infections due AMR organisms of 4.9% for AMR Gram-negative bacteria and 5.9% for MRSA [28].

Patients with AMR-associated pneumonia experienced significantly prolonged LOHS (median 14 vs. 8 days; the adjusted rate of hospitalization days was 46% higher [IRR 1.46 (95% CI 1.41–1.50)], and the adjusted odds of a stay longer than the population median was or 2.87 (95% CI 2.70–3.05), with a mean increase of six days [ATE 6.06 (95% CI 5.44–6.68)]. These findings align with prior studies: Mauldin et al. observed a significant adjusted increase in LOHS in patients with health care-associated infections (including pneumonia) due to resistant pathogens, with a 23.8% increase (*p* = 0.0003) [29] and Lat et al. also reported that patients with pneumonia caused by AMR bacteria experienced a longer LOHS (23.9 ± 23.8 days vs. 17.9 ± 22.7 days, *p* = 0.020) [30].

The effect of AMR on patient outcomes can be explained by three main determinants. First, patients infected by AMR bacteria are more likely to receive inadequate empirical antimicrobial therapy. As there is an association between delayed initiation of effective antimicrobial treatment and survival [3,6,14], this hypothesis may explain the increased mortality observed in our study. Second, some resistant bacteria are suspected to have increased virulence. The third determinant relates to host factors and comorbidities. Frail patients are at higher risk of recurrent hospitalizations and antimicrobial exposure, which, in turn, increases their risk of colonization and infection by AMR bacteria [24].

Notably, our data highlight *Staphylococcus aureus*, *Pseudomonas aeruginosa*, *Klebsiella pneumoniae*, and *Escherichia coli* as the predominant AMR pathogens, in line with surveillance data from the WHO and CDC [3,11].

### 3.1. Strengths

The strengths of this study include its large, nationally representative cohort of patients with bacterial pneumonia and etiological confirmation of the causative agent. Given the low identification rates in pneumonia cases (less than 40%, with bacterial isolates accounting for less than 14%) [16], most published studies include smaller sample sizes.

Furthermore, the use of the largest administrative database in Spain, containing standardized clinical data on hospitalized patients, ensured a comprehensive capture of clinical outcomes and risk factors, allowing for rigorous adjustment for confounders (up to eleven, such as comorbidities, VAP, age, ICU admission) using multivariate models (logistic and Poisson regression) and propensity score analyses.

We applied rigorous exclusion criteria (Figure 4) to the study sample—including the elimination of pneumonia cases without microbiological isolation, the exclusion of polymicrobial infections, and the exclusion of patients with another concomitant bacterial infection in addition to pneumonia—in order to minimize bias when comparing populations with similar clinical characteristics. Although this approach resulted in a substantial reduction in the sample size, it ensured a more homogeneous and comparable population.

To further strengthen the accuracy of our findings and avoid underestimating the prevalence of antimicrobial resistance (AMR), we restricted our analysis to data from 2017 onwards. The transition from ICD-9-CM to ICD-10-CM in 2016 introduced substantial changes in diagnostic coding that could have compromised the consistency and reliability of AMR classification. Given that 2016 represented a transitional period, with a higher likelihood of coding inconsistencies and errors, we excluded this year from the analysis to improve data quality and comparability over time.

Finally, the observed effect, despite an unbalanced population (1 AMR case vs. 18 non-AMR cases), reinforces the magnitude of the impact of AMR bacteria.

### 3.2. Limitations

*Coding biases and under-reporting (ICD-10-CM):* There is a potential risk of misclassification that may be due to ICD-10-CM coding errors or under-reporting. Nevertheless, the Spanish RAE-CMBD has been validated for research in previous studies, and such errors are partially mitigated by the large dataset size [31,32,33]. Furthermore, the volume of data analyzed contributes to minimizing the impact of occasional misclassification or under-reporting.

*Unmeasured confounders*: Key variables such as antibiotic timing or severity scores (e.g., CURB-65) were not available, which could inflate the apparent impact of AMR. However, sensitivity analyses (e.g., Poisson regression) confirmed robust effect sizes, reducing the likelihood of significant bias.

*Selection bias*: The exclusion of polymicrobial infections may have led to an underestimation of the true burden of AMR. Nonetheless, this criterion ensured a more homogeneous and clinically comparable cohort.

*Definition of AMR and appropriateness of therapy*: The definition of AMR relied on each attending physician, with variability in sampling techniques. Moreover, the database did not provide information on the adequacy or timeliness of empirical treatment, which is a major determinant of mortality. Future studies should integrate microbiological data with administrative records to refine resistance patterns and assess therapeutic adequacy.

*Competing risk bias in length of stay analyses*: The analysis of length of hospital stay (LOHS) using a Poisson regression model may be influenced by in-hospital mortality since death shortens the observation period and can bias estimates. However, it should be noted that several influential and widely cited studies have adopted similar modeling strategies, given that Poisson regression offers a simple and interpretable framework for estimating incidence rate ratios (IRR) [34].

Finally, selection bias is inherent in the retrospective design. To minimize this, our group previously validated the EVCI as the best predictor of in-hospital mortality in our population, to eliminate the confounding effect caused by the patients’ different levels of comorbidity [35]. In addition, we applied strict exclusion criteria (e.g., polymicrobial bacterial pneumonia, other bacterial infections) to obtain a more reproducible and comparable cohort.

### 3.3. Conclusions

Consistent with previous evidence, our findings demonstrate that antimicrobial-resistant (AMR) is associated with longer hospital stays and an up to 50% higher risk of mortality, emphasizing the urgent need for action.

Despite the implementation of control policies in place over the past decade, these alarming data highlight the persistence of the problem and compel policymakers to strengthen AMR surveillance and ensure adequate resource allocation in order to mitigate its clinical and economic impact. Clinicians, in turn, must reinforce antimicrobial stewardship to limit the emergence of resistant infections and incorporate rapid diagnostic tools to minimize the impact of AMR on patient outcomes.

### 3.4. Key Recommendations and Future Directions

Future research should investigate region-specific AMR trends to guide local antimicrobial stewardship programs, explore cost-effectiveness of rapid diagnostics for reducing AMR-related mortality, and assess long-term outcomes (e.g., post-discharge survival).

The presence of AMR bacteria complicates the clinical management of infections [11]. Our findings advocate for multifaceted interventions:

Optimization of antimicrobial use: Personalized empirical therapy guided by local resistance patterns.

Improved diagnosis: Rapid molecular testing to enable the early identification of AMR pathogens and the timely initiation of targeted therapy.

Strengthening antimicrobial stewardship programs: This will mitigate the natural progression of resistance.

Investment in nosocomial infection control measures: This is particularly important for high-risk pathogens such as Staphylococcus aureus, Pseudomonas aeruginosa, Escherichia coli, and Klebsiella pneumoniae.

In Spain, the implementation of the “Zero Resistance” project recommendations has been associated with a significant reduction in infections caused by multidrug-resistant bacteria acquired during ICU stays [IRR 0.67 (95% CI 0.57–0.80)] [34].

Future research should investigate region-specific AMR trends to inform local antimicrobial stewardship programs, evaluate the cost-effectiveness of rapid diagnostics in reducing AMR-related mortality, and assess long-term patient outcomes, including post-discharge survival.

## 4. Materials and Methods

### 4.1. Study Design

A retrospective cohort study was conducted on adult patients hospitalized with bacterial pneumonia and discharged from hospitals within the Spanish National Health System between 2017 and 2022, using data from the RAE-CMBD administrative database. Hospitalization episodes with antimicrobial resistance bacteria—identified through ICD-10-CM resistance codes (Z16.1, Z16.2)—were compared with hospitalization episodes without these codes.

### 4.2. Patients

Adult patients admitted to public or private hospitals with a diagnosis of bacterial pneumonia were eligible. Patients were followed until discharge.

Key exclusion criteria were as follows:All patients under 18 years of age.All cases of pneumonia due to bacteria from the *Mycobacteriales* and *Spirochaetales* orders.All patients with bacterial pneumonia and any other concurrent bacterial infectious disease.All cases coded with bacterial pneumonia attributed to more than one causative agent.All cases coded with a diagnosis of bacterial pneumonia in which the causative agent was not specified or not identified.Patients codified with diagnosis codes for very unusual bacterial pneumonias, e.g., pneumonia due to typhoidal salmonella, pneumonia due to non-typhoidal Salmonella pneumonia, pneumonic plague, glanders, and melioidosis.

Appendix A detail the ICD-10-CM exclusion codes applied to the study cohort to select eligible patients.

### 4.3. Procedures

The RAE-CMBD is the largest administrative database maintained by the Spanish ministry of health, containing the standardized clinical data of hospitalized patients. This registry contains, for each hospitalization, the coding of the main diagnosis (cause of the hospital admission) and up to 19 secondary diagnoses (other conditions that influence the patient’s clinical course), coded according to the ICD-10-CM, Spanish version [14].

The following variables were collected from the RAE-CMBD:Administrative variables: Year of the registry, center, and anonymized patient ID number.Sociodemographic variables: Date of birth, sex (male/female/not specified), and country of birth. Sex data are collected in RAE-CMBD as recorded in the patient’s medical history during admission according to the patient’s self-reported information.Clinical variables: Date of hospital admission, date of discharge, type of discharge (death/home/transfer to another hospital/voluntary discharge/transfer to social-health center), intensive care unit (ICU) admission (yes/no), length of ICU stay (in days), main diagnosis (ICD-10-CM), secondary diagnoses (ICD-10-CM), procedures 1 to 20 (ICD-10-CM), all patient-refined diagnosis-related groups (APR-DRG) severity level (null/minor/moderate/major/extreme), and APR-DRG mortality risk level (low/medium/high/extreme).

### 4.4. Outcomes

Mortality was defined as the number of hospitalization episodes in which death was the reason for hospital discharge. LOHS was defined as the number of days between the discharge date and the hospital admission date.

The outcomes were evaluated in both the AMR group and the non-AMR based on the final cohort of eligible hospitalization episodes, as detailed in the following flow chart (Figure 4).

### 4.5. Statistical Analysis

First, a univariate regression model studied the risk factors for in-hospital mortality in patients hospitalized with bacterial pneumonia. Comorbidity at each hospitalization episode was assessed using the Elixhauser–van Walraven Comorbidity Index (EVCI). Appendix A shows the calculation of the EVCI. Appendix A details the ICD-10-CM codes used to study the risk factors. Subsequently, multivariate logistic regression models were built, as primary analyses, including the AMR bacteria label, with appropriate covariates based on the risk factors.

We used a phylogenetic structural classification (Gram-negative and Gram-positive) rather than an anatomical one (respiratory vs. enteric microorganisms), which would have been more appropriate for pneumonia etiology since the primary goal was to assess the impact of microbial resistance on health outcomes, regardless of the infection source.

To apply logistic regression, the EVCI score, age at admission (years), and LOHS (days) were categorized into quintiles based on their distributions (stratum 1: quintile 1; strata 2: quintile 2; strata 3: quintile 3; strata 4: quintile 4; and strata 5: quintile 5). This approach enables a clearer comparison of the influence of each stratum (quintile) on the outcome variable (in-hospital mortality) as this relationship is non-linear (see Appendix A). Similar methods have been reported by other authors to yield more reliable results [31,35]. The stratum associated with lower mortality was chosen as the reference category: quintile 1 in age and EVCI and quintile 2 in LOHS. In logistic regression, the dependent effect of LOHS was measured as the probability of a prolonged hospitalization, using the population median (8 days) as the cut-off point.

### 4.6. Sensitivity Analysis

As a secondary method, we tested the hypotheses using the same covariates as in the original analysis using a modified Poisson multivariate regression model with robust standard errors, which provided the incidence rate ratio (IRR).

Also, propensity score matching tests without replacement (matching ratio: 1:1 exact; caliper of 0.1) were performed for both the entire population (average treatment effect (ATE)) and for the AMR bacterial group (average treatment effect on the treated (ATET)).

Statistical analysis was performed using Stata/MP v17.

## Figures and Tables

**Figure 1 antibiotics-14-01006-f001:**
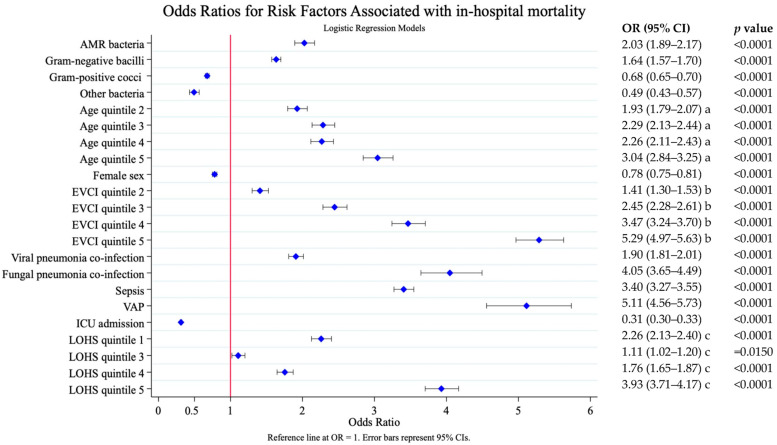
Risk factors associated with in-hospital mortality. Results of the univariate logistic analysis. AMR, antimicrobial resistance; EVCI, Elixhauser–van Walraven comorbidity index; VAP, ventilator-associated pneumonia; ICU, intensive care unit. LOHS, length of hospital stay. Age (years)—quintile 1: 18–56; quintile 2: 57–68; quintile 3: 69–77; quintile 4: 78–85; and quintile 5: 86–117. EVCI—quintile 1: −18 to 0; quintile 2: 1–4; quintile 3: 5–8; quintile 4: 9–13; and quintile 5: 14–48. Length of stay (days): quintile 1: 0–4; quintile 2: 5–7; quintile 3: 8–9; quintile 4: 10–16; and quintile 5: 17–776. a. Reference age quintile; b. Reference EVCI quintile 1; c. Reference LOHS quintile 2.

**Figure 2 antibiotics-14-01006-f002:**
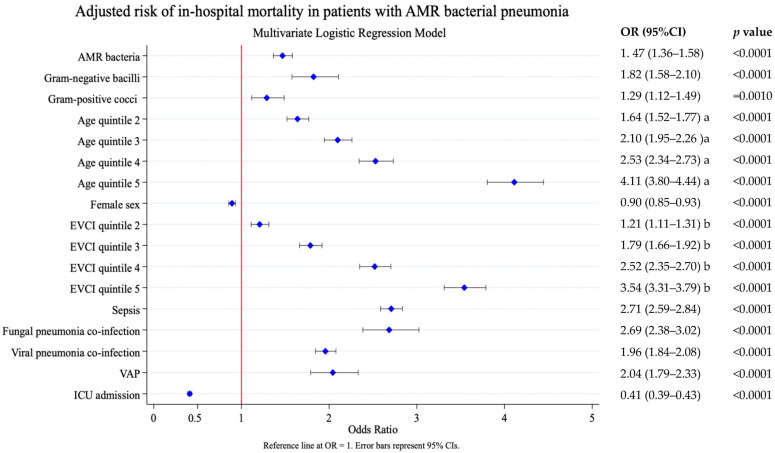
Risk factors associated with in-hospital mortality. Results of the multivariate logistic analysis adjusted by the following covariates: AMR bacterial label, pneumonia due to Gram-positive cocci, pneumonia due to Gram-negative bacilli, age at admission, sex, comorbidity (EVCI), fungal pneumonia co-infection, viral pneumonia co-infection, sepsis, ventilator-associated pneumonia, and ICU admission. AMR, antimicrobial resistance; EVCI, Elixhauser–van Walraven comorbidity index; VAP, ventilator-associated pneumonia; ICU, intensive care unit. Age at admission (years)—quintile 1: 18–56; quintile 2: 57–68; quintile 3: 69–77; quintile 4: 78–85; and quintile 5: 86–117. EVCI—quintile 1: −18 to 0; quintile 2: 1–4; quintile 3: 5–8; quintile 4: 9–13; and quintile 5: 14–48. Length of stay (days)—quintile 1: 0–4; quintile 2: 5–7; quintile 3: 8–9; quintile 4: 10–16; and quintile 5: 17–776. a. Reference to age quintile 1; b. Reference EVCI quintile 1.

**Figure 3 antibiotics-14-01006-f003:**
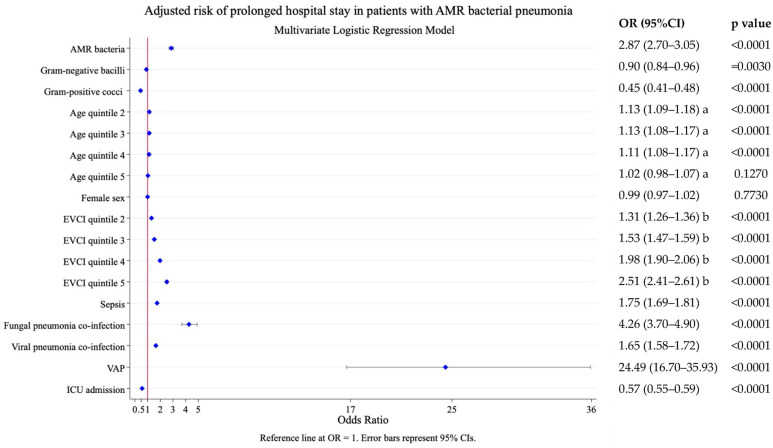
Risk factors associated with prolonged length of hospital stay (i.e., stays longer than 8 days, the median stay of the population). Results of the multivariate logistic analysis adjusted by the following covariates: AMR bacterial label, age, sex, comorbidity (EVCI), sepsis, fungal pneumonia co-infection, viral pneumonia co-infection, VAP, and ICU admission. AMR, antimicrobial resistance; EVCI, Elixhauser–van Walraven comorbidity index; VAP, ventilator-associated pneumonia; ICU, intensive care unit. Age at admission (years)—quintile 1: 18–56; quintile 2: 57–68; quintile 3: 69–77; quintile 4: 78–85; and quintile 5: 86–117. EVCI—quintile 1: −18 to 0; quintile 2: 1–4; quintile 3: 5–8; quintile 4: 9–13; and quintile 5: 14–48. a. Reference to age quintile 1; b. Reference EVCI quintile 1.

**Figure 4 antibiotics-14-01006-f004:**
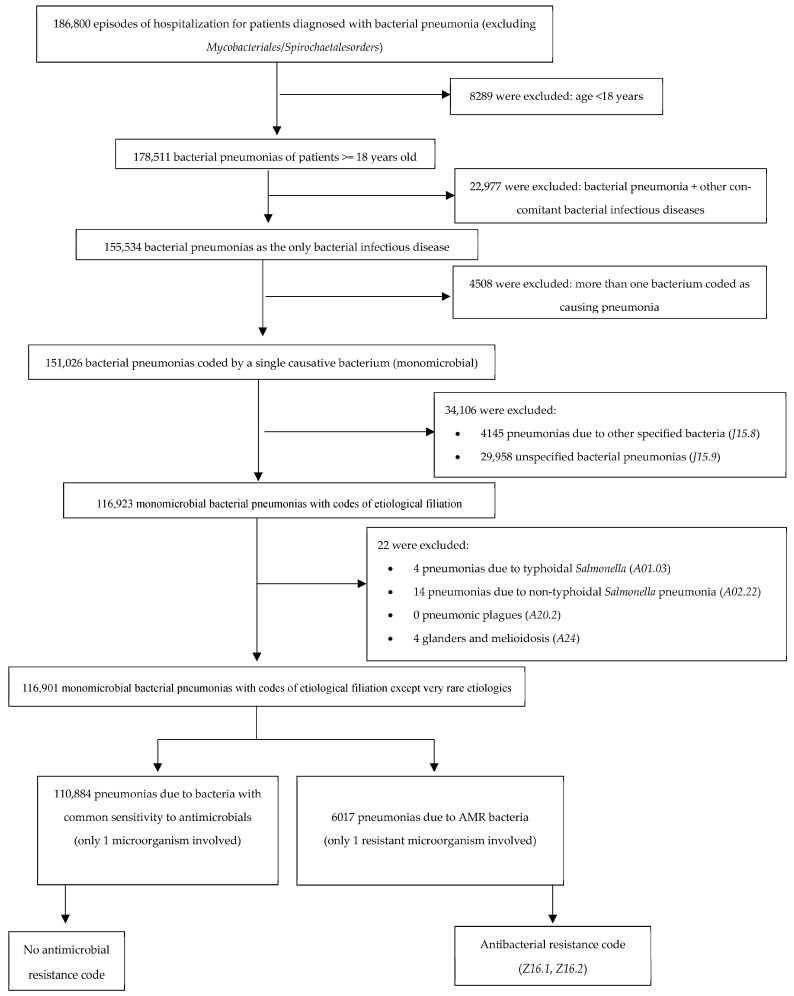
Selection of study population. Number of potentially eligible participants, algorithm for determining their eligibility, and those confirmed as eligible for the study. AMR, antimicrobial resistance.

**Table 1 antibiotics-14-01006-t001:** Prevalence of antimicrobial drug resistance (AMR) across the study years.

Fiscal Year	Total Cases	AMR Cases	Prevalence (%)
2017	17,561	1028	5.85
2018	21,462	1089	5.07
2019	22,930	1121	4.89
2020	19,098	935	4.90
2021	15,016	868	5.78
2022	20,834	976	4.68

**Table 2 antibiotics-14-01006-t002:** Demographic and clinical characteristics, length of stay, and in-hospital mortality in hospitalized adults with and without an AMR bacteria code.

Variable	Bacterial Pneumonia Hospitalizations Without AMR Bacteria Code (n = 110,884)	Bacterial Pneumonia Hospitalizations With AMR Bacteria Code (n = 6017)	*p*-Value
Age (years), median (IQR)	72 (59–83)	75 (64–84)	<0.0001 ^a^
Sex, n (%)			
Men	69,255 (62.46)	4255 (70.72)	<0.0001 ^b^
Women	41,628 (37.54)	1762 (29.28)	<0.0001 ^b^
EVCI score, median (IQR)	5 (0–12)	8 (3–15)	<0.0001 ^a^
Severity of illness APR-DRG, n (%)			
Null	84 (0.08)	18 (0.13)	<0.0001 ^b^
Minor	11,160 (10.06)	155 (2.58)	<0.0001 ^b^
Moderate	26,582 (23.97)	808 (13.43)	<0.0001 ^b^
Major	46,086 (41.56)	2415 (40.14)	<0.0001 ^b^
Extreme	26,972 (24.32)	2631 (43.73)	<0.0001 ^b^
Risk of mortality APR-DRG, n (%)			
Null	84 (0.08)	8 (0.13)	<0.0001 ^b^
Low	18,830 (16.98)	293 (4.87)	<0.0001 ^b^
Medium	27,043 (24.39)	1258 (20.91)	<0.0001 ^b^
High	43,363 (39.11)	2643 (43.93)	0.2910 ^b^
Extreme	21,564 (19.45)	1815 (30.16)	<0.0001 ^b^
Viral pneumonia co-infection, n (%)	10,547 (9.51)	597 (9.92)	<0.0001 ^b^
Fungal pneumonia co-infection, n (%)	1456 (1.31)	226 (3.76)	<0.0001 ^b^
Sepsis, n (%)	18,590 (16.77)	1297 (21.56)	<0.0001 ^b^
ICU admission, n (%)	15,692 (14.15)	1336 (22.20)	<0.0001 ^b^
Mechanical ventilation, n (%)	8785 (7.92)	964 (16.02)	<0.0001 ^b^
Ventilator-associated pneumonia, n (%)	1108 (1.00)	180 (2.99)	<0.0001 ^b^
Extracorporeal membrane oxygenation, n (%)	221 (0.20)	24 (0.40)	<0.0010 ^b^
Nosocomial condition	3164 (2.85)	628 (10.44)	<0.0001 ^b^
Length of stay (days), median (IQR)	8 (5–13)	14 (8–24)	<0.0001 ^a^
ICU length of stay (days), median (IQR)	8 (3–17)	13 (5–26)	<0.0001 ^a^
In-hospital mortality, n (%)	11,147 (10.05)	1111 (18.46)	<0.0001 ^b^

IQR, interquartile range; EVCI, Elixhauser–van Walraven comorbidity index; APR-DRG, all patient-refined diagnosis-related groups (APR-DRG); ICU, intensive care unit. ^a^ Mann–Whitney U test (Prob > z); ^b^ χ2 test.

**Table 3 antibiotics-14-01006-t003:** Microbial etiology of the bacterial pneumonia according to International Classification of Diseases 10th Revision Clinical Modification (ICD-10-CM).

Microbial Etiology	Non-AMR Code, n (%)	AMR Code, n (%)	Total Bacterial Etiology, n (%)	Total AMR Bacterial, n (%)
Gram-positive cocci group				
*Streptococcus pneumoniae*	69,697 (99.13)	615 (0.87)	70,312 (60.15)	615 (10.22)
*Streptococcus* group B	136 (94.44)	8 (5.56)	144 (0.12)	8 (0.13)
*Streptococcus*, other	1821 (97.59)	45 (2.41)	1866 (1.60)	45 (0.75)
*Staphylococcus aureus*	3509 (56.72)	2678 (43.28)	6187 (5.29)	2,678 (44.52)
*Staphylococcus*, other	794 (91.58)	73 (8.42)	867 (0.74)	73 (1.21)
*Staphylococcus*, unspecified	288 (92.60)	23 (7.40)	311 (0.27)	23 (0.38)
Total Gram-positive cocci group	76,245 (95.68)	3442 (4.32)	79,687 (68.17)	3442 (57.20)
Gram-negative bacilli group				
*Pseudomonas aeruginosa*	7438 (89.65)	859 (10.35)	8297 (7.10)	859 (14.28)
*Haemophilus influenzae*	5430 (96.36)	205 (3.64)	5635 (4,82)	205 (3.41)
*Klebsiella pneumoniae*	3321 (85.61)	558 (14.39)	3879 (3.32)	558 (9.27)
*Escherichia coli*	2560 (84.80)	459 (15.20)	3019 (2.58)	459 (7.63)
Gram-negative aerobic bacteria, other	4441 (91.62)	406 (8.38)	4847 (4.15)	406 (6.75)
*Bordetella pertussis/Bordetella parapertussis* (Whooping cough)	27 (93.10)	2 (6.90)	29 (0.02)	2 (0.03)
Total Gram-negative bacilli group	30,673 (92.42)	2514 (7.58)	33,187 (28.39)	2514 (41.78)
Other bacteria group				
*Legionella pneumophila*	7456 (99.67)	25 (0.33)	7481 (6.40)	25 (0.42)
*Actinomyces* spp. (Pulmonary actinomycosis)	264 (97.06)	8 (2.94)	272(0.23)	8 (0.13)
*Mycoplasma pneumoniae*	1444 (99.59)	6 (0.41)	1450 (1.24)	6 (0.10)
*Chlamydophila pneumoniae*	1095 (99.10)	10 (0.90)	1105 (0.95)	10 (0.17)
*Nocardia* spp. (Pulmonary nocardiosis)	635 (95.49)	30 (4.51)	665 (0.57)	30 (0.50)
*Coxiella burnetii* (Q fever)	461 (98.72)	6 (1.28)	467 (0.40)	6 (0.10)
*Brucella* spp. (Brucellosis)	33 (97.06)	1 (2.94)	34 (0.03)	1 (0.02)
*Neisseria gonorrhoeae* (Gonococcal pneumonia)	21 (100.00)	0 (0.00)	21 (0.02)	0 (0.00)
*Francisella tularensis* (Pulmonary tularaemia)	12 (100.00)	0 (0.00)	12 (0.01)	0 (0.00)
*Bacillus anthracis* (Pulmonary anthrax)	1 (100.00)	0 (0.00)	1 (0.00)	0 (0.00)
Total other bacteria group	3966 (98.49)	61 (1.51)	4027 (3.44)	61 (1.01)
Total bacteria	110,884 (100.00)	6,017 (100.00)	116,901 (100.00)	6017 (100.00)

**Table 4 antibiotics-14-01006-t004:** Results by subgroup (type of bacteria) of the multivariate logistic analysis for hospital mortality adjusted by the following covariates: age at admission, sex, comorbidity (EVCI), fungal pneumonia co-infection, viral pneumonia co-infection, sepsis, ventilator-associated pneumonia, and ICU admission.

	Adjusted OR	95% CI, Lower Limit	95% CI, Upper Limit	*p* > |z|
Pneumonia due AMR GPC	1.62	1.48	1.78	<0.0001
Pneumonia due to AMR GNB	1.41	1.26	1.58	<0.0001
Pneumonia due to other AMR bacteria	0.70	0.27	1.81	=0.4650
Age at admission quintile 2 a	1.68	1.55	1.80	<0.0001
Age at admission quintile 3 a	2.14	1.99	2.31	<0.0001
Age at admission quintile 4 a	2.55	2.36	2.75	<0.0001
Age at admission quintile 5 a	4.06	3.76	4.39	<0.0001
Female sex	0.87	0.83	0.90	<0.0001
EVCI quintile 2 b	1.22	1.13	1.33	<0.0001
EVCI quintile 3 b	1.81	1.69	1.95	<0.0001
EVCI quintile 4 b	2.57	2.39	2.76	<0.0001
EVCI quintile 5 b	3.64	3.41	3.90	<0.0001
Fungal pneumonia co-infection	2.85	2.53	3.21	<0.0001
Viral pneumonia co-infection	1.91	1.80	2.03	<0.0001
Sepsis	2.64	2.52	2.76	<0.0001
VAP	2.22	1.95	2.54	<0.0001
ICU admission	0.39	0.37	0.41	<0.0001

AMR, antimicrobial resistance; GPC, Gram-positive cocci; GNB, Gram-negative bacilli; EVCI, Elixhauser–van Walraven comorbidity index; VAP, ventilator-associated pneumonia; ICU, intensive care unit; LOHS, length of hospital stay. Age at admission (years)—quintile 1: 18–56; quintile 2: 57–68; quintile 3: 69–77; quintile 4: 78–85; and quintile 5: 86–117. EVCI—quintile 1: −18 to 0; quintile 2: 1–4; quintile 3: 5–8; quintile 4: 9–13; and quintile 5: 14–48. a. Reference to age quintile1; b. Reference EVCI quintile 1.

**Table 5 antibiotics-14-01006-t005:** Sensitivity analysis: Poisson regression: analysis adjusted for in-hospital mortality and LOHS by the following covariates: AMR bacterial label, pneumonia due to Gram-positive cocci, pneumonia due to Gram-negative bacilli, age at admission, sex, comorbidity (EVCI), fungal pneumonia co-infection, viral pneumonia co-infection, sepsis, ventilator-associated pneumonia, ICU admission and in-hospital mortality.

		In-Hospital Mortality					Length of Hospital Stay			
	Adj. IRR	Robust Std. Err	95% CI, Lower Limit	95% CI, Upper Limit	*p* > |z|	Adj. IRR	Robust Std. Err.	95% CI, Lower Limit	95% CI, Upper Limit	*p* > |z|
AMR Bacterial	1.34	0.0374	1.26	1.41	<0.0001	1.46	0.0220	1.41	1.50	<0.0001
Pneumonia due GPC	1.25	0.0777	1.11	1.42	<0.0001	0.74	0.0106	0.72	0.76	<0.0001
Pneumonia due to GNB	1.65	0.1034	1.46	1.86	0.0010	1.10	0.0179	1.06	1.13	<0.0001
Age at admission quintile 2 a	1.49	0.0479	1.40	1.59	<0.0001	1.00	0.0152	0.98	1.04	=0.7220
Age at admission quintile 3 a	1.80	0.0577	1.70	1.93	<0.0001	0.97	0.0147	0.94	1.00	0.0260
Age at admission quintile 4 a	2.13	0.0734	2.00	2.29	<0.0001	0.90	0.0143	0.87	0.93	<0.0001
Age at admission quintile 5 a	3.23	0.1192	3.00	3.47	<0.0001	0.83	0.0129	0.80	0.85	<0.0001
Female sex	0.91	0.0163	0.88	0.94	<0.0001	0.97	0.0072	0.95	0.98	<0.0001
EVCI quintile 2 b	1.20	0.0454	1.11	1.30	<0.0001	1.11	0.0143	1.08	1.14	<0.0001
EVCI quintile 3 b	1.70	0.0557	1.60	1.81	<0.0001	1.22	0.0161	1.19	1.26	<0.0001
EVCI quintile 4 b	2.26	0.0727	2.12	2.41	<0.0001	1.34	0.0205	1.30	1.38	<0.0001
EVCI quintile 5 b	2.92	0.0902	2.75	3.11	<0.0001	1.43	0.0177	1.40	1.47	<0.0001
Fungal pneumonia co-infection	1.83	0.0702	1.70	1.97	<0.0001	1.52	0.0376	1.45	1.60	<0.0001
Viral pneumonia co-infection	1.65	0.0378	1.57	1.72	<0.0001	1.20	0.0131	1.18	1.23	<0.0001
Sepsis	2.17	0.0494	2.07	2.27	<0.0001	1.29	0.0162	1.26	1.32	<0.0001
VAP	1.49	0.0649	1.36	1.62	<0.0001	2.11	0.0558	2.00	2.22	<0.0001
ICU admission	0.49	0.0228	0.45	0.53	<0.0001	0.68	0.0135	0.66	0.70	<0.0001

Adj. adjusted; std. err. standard error; IRR, incidence rate ratio; CI, confidence interval; AMR, antimicrobial resistance; GPC, Gram-positive cocci; GNB, Gram-negative bacilli; EVCI, Elixhauser–van Walraven comorbidity index; VAP, ventilator-associated pneumonia; ICU, intensive care unit; LOHS, length of hospital stay. Age at admission (years)—quintile 1: 18–56; quintile 2: 57–68; quintile 3: 69–77; quintile 4: 78–85; and quintile 5: 86–117. EVCI—quintile 1 (reference): −18 to 0; quintile 2: 1–4; quintile 3: 5–8; quintile 4: 9–13; and quintile 5: 14–48. a. Reference to age quintile change; b. Reference EVCI quintile change.

**Table 6 antibiotics-14-01006-t006:** Sensitivity analysis: Propensity score matching test of AMR bacterial by the following covariates: pneumonia due to Gram-positive cocci group, pneumonia due to Gram-negative bacilli group, age (quintile), sex, Elixhauser–van Walraven comorbidity index (quintile), viral pneumonia co-infection, fungal pneumonia co-infection, sepsis, ventilator-associated pneumonia, and ICU admission.

	Coefficient	Std. Err.	95% CI, Lower Limit	95% CI, Upper Limit	*p* > |z|
In-hospital mortality					
ATET, %	4.01	0.0051	3.02	5.00	<0.0001
ATE, %	3.82	0.0048	2.87	4.77	<0.0001
Length of hospital stay					
ATET, days	5.93	0.2975	5.34	6.51	<0.0001
ATE, days	6.06	0.3155	5.44	6.68	<0.0001

ATET, average treatment effect on the treated; ATE, average treatment effect; std. err., standard error; CI, confidence interval. Results are referred to as Case vs. Control, caliper of 0.1; Match ratio: 1:1 exact.

## Data Availability

The database used (RAE-CMBD) was provided by the Spanish Ministry of Health. Access to these data is public, and any author may request access to the anonymized data following the instructions of the Spanish government through the following website: https://www.sanidad.gob.es/estadEstudios/estadisticas/estadisticas/estMinisterio/SolicitudCMBD.htm.

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
