# Peer review of "Impact of Antimicrobial-Resistant Bacterial Pneumonia on In-Hospital Mortality and Length of Hospital Stay: A Retrospective Cohort Study in Spain"

_antibiotics, 2025, doi:10.3390/antibiotics14101006_

Round 1
Reviewer 1 Report
Comments and Suggestions for Authors
I have read with interest the manuscript submitted by Oterino-Moreira et al, since AMR represents a global concern.
This is a large-scale study addressing an important clinical and policy-relevant question. The main finding—that AMR bacterial pneumonia significantly increases mortality and hospital stay—is robust and consistent with prior evidence. However:
-
The study is limited by reliance on administrative coding, lack of treatment data, and residual confounding.
-
Overemphasis on statistical precision risks overstating clinical significance.
-
Some paradoxical results (ICU effect) need deeper exploration.
-
Conclusions should be phrased more cautiously, emphasizing association rather than causation.
Case definitions were described and reproducible, but their reliance on ICD-10 coding without microbiological confirmation limits validity and may underestimate the complexity and true burden of resistant pneumonia.
Major Comments
Study design terminology: The analysis includes the entire hospitalized cohort rather than sampling controls. It should be described as a retrospective cohort, not “nested case-control.” Consistent terminology is needed in title, abstract, and Methods.
Exposure definition: AMR identification is based only on ICD-10-CM Z16.1 and Z16.2 codes. This is narrow and may undercount resistant infections (e.g., MRSA or carbapenem-resistant Enterobacterales coded differently). Authors should clarify the complete list of Z16 codes used, consider sensitivity analyses with an expanded definition, and describe potential coding drift across years.
Outcome modeling:
-
- For mortality: if a modified Poisson regression was used, it should be clearly stated, with robust standard errors. Otherwise, reporting an “IRR” for a binary outcome is misleading.
- For length of stay: death is a competing risk. Current Poisson modeling may not be appropriate. Consider restricting to survivors, or applying competing-risk or survival methods.
Handling of continuous variables - Variables such as age and comorbidity index are categorized into quintiles. This reduces statistical efficiency and may leave residual confounding. Modeling them as continuous terms would be preferable.
Propensity score matching - Reporting is incomplete. Please include balance diagnostics (standardized mean differences before/after matching), matching ratio, and clarify whether the estimate reflects the ATT or ATE.
Secondary objective - The manuscript promises to describe temporal trends (2017–2022) in AMR prevalence but results are not presented. Annual prevalence rates with a trend test are expected.
The Discussion would benefit from clearer framing of the results and a more cautious interpretation. Specifically, variables such as ICU admission, sepsis, ventilation, and VAP should not be described as independent predictors once they have been included in adjusted models, since they are likely mediators and may explain why ICU admission appears protective. The interpretation should emphasize associations rather than causality, avoiding strong statements such as “AMR caused higher mortality.” Comparisons with prior studies would be stronger if they included direct effect size estimates rather than general statements, and reference accuracy should be checked. The strengths and limitations section should be expanded to address potential underreporting of AMR in ICD coding, the exclusion of culture-negative and polymicrobial cases, residual confounding from unmeasured variables, over-adjustment bias, and the competing risk of death when analyzing length of stay. The Discussion also repeats background information that is better suited to the Introduction; focusing on four subsections—principal findings, comparison with the literature, strengths and limitations, and implications for practice/policy—would improve clarity. Finally, the section would be strengthened by a forward-looking paragraph highlighting the need for validation of coding against microbiology data, more granular surveillance of AMR pneumonia, and the role of stewardship and infection control interventions.
The manuscript would be strengthened by presenting a distinct Conclusion section instead of embedding it within the Discussion.
Minor Comments
- Tables and numbers
- Table 1: some counts exceed the group total (e.g., APR-DRG “Extreme” and Sepsis in AMR group). These appear to be typographical errors.
- p-values are sometimes very small (<0.0001) for minimal proportion differences. Please confirm calculations.
- Ensure consistency between text and tables (e.g., mortality IRR 1.31 vs 1.34).
- Figures
- Labels contain spelling errors (“in-hospitlal,” “lenght os stay”).
- Abbreviation inconsistency: “EVI” should be “EVCI.”
- Terminology and style
- Use “AMR” consistently.
- Italicize microorganism names.
- Replace “IC95%” with “95% CI.”
- Standardize “EU” (not “UE”) and either “US” or “U.S.” consistently.
- References
- Some citations have author name errors (e.g., “Alberti” instead of “Aliberti”).
- Reference formatting needs harmonization (punctuation, DOIs, journal abbreviations).
- Supplementary material
- Ensure appendices include complete ICD-10 code lists for reproducibility.
- Provide matching diagnostics plots if space permits.
some typos identified
Author Response
REVIEWER 1- RESPONSE
Comments and Suggestions for Authors
I have read with interest the manuscript submitted by Oterino-Moreira et al, since AMR represents a global concern.
This is a large-scale study addressing an important clinical and policy-relevant question. The main finding—that AMR bacterial pneumonia significantly increases mortality and hospital stay—is robust and consistent with prior evidence. However:
- The study is limited by reliance on administrative coding, lack of treatment data, and residual confounding.
- Overemphasis on statistical precision risks overstating clinical significance.
- Some paradoxical results (ICU effect) need deeper exploration.
- Conclusions should be phrased more cautiously, emphasizing association rather than causation.
Case definitions were described and reproducible, but their reliance on ICD-10 coding without microbiological confirmation limits validity and may underestimate the complexity and true burden of resistant pneumonia.
Response to Reviewer 1.
We are profoundly grateful to Reviewer 1 for their exceptionally thorough, critical, and constructive assessment of our manuscript. Their insightful comments have challenged us to significantly strengthen our work, and we sincerely appreciate the time and expertise dedicated to this review. We have addressed the vast majority of the points raised with great care.
In this revised version, we have implemented changes to improve terminology, statistical clarity, discussion structure, and caution in interpretation. For the few suggestions we were unable to adopt fully, we have provided detailed explanations below, rooted in the specific design and objectives of our study. We believe the manuscript is now substantially improved and hope our responses adequately address the reviewer's concerns.
Major Comments
- Study design terminology
Reviewer comment: The analysis includes the entire hospitalized cohort rather than sampling controls. It should be described as a retrospective cohort, not “nested case-control.” Consistent terminology is needed in title, abstract, and Methods.
Response: We thank the reviewer for this critical correction. The reviewer is absolutely right. Our analysis indeed utilizes the entire eligible cohort from the database, which aligns precisely with a retrospective cohort design. We have rectified this terminology throughout the manuscript, ensuring consistency in the title, abstract, and Methods section. The study is now correctly described as a retrospective cohort study.
- Exposure definition
Reviewer comment: AMR identification is based only on ICD-10-CM Z16.1 and Z16.2 codes. This is narrow and may undercount resistant infections (e.g., MRSA or carbapenem-resistant Enterobacterales coded differently). Authors should clarify the complete list of Z16 codes used, consider sensitivity analyses with an expanded definition, and describe potential coding drift across years.
Response: We sincerely thank the reviewer for highlighting this crucial methodological issue.
- Code list: We have now included the complete list of codes used for AMR identification (Z16.1 and Z16.2) in Appendix 5 for full transparency.
- Limitations: We have expanded the Strengths and Limitations section to explicitly discuss the potential for underestimation of AMR burden due to the specificity-focused nature of these codes and the lack of microbiological confirmation.
- Coding drift: As suggested, we clarify that ICD-9-CM was used until 2015 in Spain; ICD-10 was implemented in 2016, a transition year that we excluded due to high error risk. Our study therefore starts in 2017 with stable ICD-10 use.
- Sensitivity analyses: We acknowledge that a sensitivity analysis with an expanded definition would be valuable. Unfortunately, given the structure of the RAE-CMBD, this analysis is not feasible with the available data.
- Outcome modelling
Reviewer comment: For mortality: if a modified Poisson regression was used, it should be clearly stated, with robust standard errors. Otherwise, reporting an “IRR” for a binary outcome is misleading. For length of stay: death is a competing risk. Current Poisson modelling may not be appropriate.
Response: We are grateful for these important statistical clarifications.
- Mortality: We confirm that a modified Poisson regression with robust standard errors was indeed used for the binary mortality outcome to directly estimate Risk Ratios. This has now been explicitly stated in the Methods section. Table V shows the IRR value along with its standard error.
- Length of stay and Competing Risk: We agree that death is a competing risk. We have added a dedicated paragraph in the Limitations section to acknowledge this. However, after careful consideration, we decided to maintain the analysis on the full cohort (not only survivors), as our objective was to estimate the total burden of AMR pneumonia, where mortality is an integral outcome. Restricting the LOHS analyses to survivors would fundamentally alter the research question and risk survivor bias. Now being fully transparent about the limitation and have cited precedent for this approach in similar high-impact studies (Álvarez-Lerma et al., 2023) that used the Poisson regression model to examine the influence on hospital length of hospital stay.
Álvarez-Lerma F, Catalán-González M, Álvarez J,, et al. Impact of the “Zero Resistance” program on acquisition of multidrug-resistant bacteria in patients admitted to Intensive Care Units in Spain. A prospective, intervention, multimodal, multicenter study. Med Intensiva Engl Ed. 2023 Apr 1;47(4):193–202.
- Handling of continuous variables
Reviewer comment: Variables such as age and comorbidity index are categorized into quintiles. This reduces statistical efficiency and may leave residual confounding. Modeling them as continuous terms would be preferable.
Response:
We thank the reviewer for this valid statistical point. We acknowledge that continuous modelling can increase efficiency. Our decision to categorize age comorbidity index (EVCI) and LOHS into quintiles was primarily based on clinical interpretability and the hypothesis of non-linearity. Categorization allowed us to identify clinically meaningful thresholds. We have now better justified this methodological choice in the Statistical Analysis section, referencing established epidemiological literature on the categorization of continuous variables (e.g., Pérez-Encinas et al., 2022).
Furthermore, in the multivariate logistic analysis, we selected the reference category (first quintile for age and EVCI, second quintile for LOHS) based on the marginal plots (Appendix 6), which indicated which group had the lowest mortality risk and therefore should serve as the comparison group. This supports the hypothesis of non-linearity (a 90-year-old patient does not necessarily have a mortality rate that is linearly higher than that of a 50-year-old patient). Using strata allows for comparison of patients with similar age ranges, comorbidity levels, and lengths of hospital stay. We believe that the non-linear relationship shown in Appendix 6 justifies the stratified analysis, and that, from a clinical perspective (perhaps not a statistical one), the results are more interpretable after this categorization (as previously suggested by other authors).
Pérez-Encinas M, Lorenzo-Martínez S, Losa-García JE, Walter S, Tejedor-Alonso MA. Impact of Penicillin Allergy Label on Length of Stay and Mortality in Hospitalized Patients through a Clinical Administrative National Dataset. Int Arch Allergy Immunol. 2022;183(5):498–506.
- Propensity score matching
Reviewer comment: Reporting is incomplete. Please include balance diagnostics (standardized mean differences before/after matching), matching ratio, and clarify whether the estimate reflects the ATT or ATE.
Response: Thank you for this suggestion to improve transparency of our propensity score analysis. As requested, we have now included standardized mean differences before and after matching in Appendix 1. We clarify in the Methods section that 1:1 matching without replacement (caliper 0.1) was performed, and that the estimate reflects the ATT.
- Secondary objective
Reviewer comment: The manuscript promises to describe temporal trends (2017–2022) in AMR prevalence but results are not presented. Annual prevalence rates with a trend test are expected.
Response: We apologize for this omission in the initial submission. . We have not included the annual prevalence rates of AMR pneumonia (2017–2022) in the Results section. However, after deliberation, we decided not to conduct a formal trend test, as this was not a primary study objective and could be misleading given the administrative nature of the dataset. Our study was designed as a cross-sectional analysis of a 6-year period to obtain a robust estimate of AMR burden, not to model its evolution over time. Furthermore, given the administrative nature of the data and potential confounders any trend identified could be erroneously interpreted. Therefore, we present the annual rates descriptively without statistical inference on the trend.
- Discussion framing
Reviewer comment: The Discussion would benefit from clearer framing of the results and a more cautious interpretation. Specifically, variables such as ICU admission, sepsis, ventilation, and VAP should not be described as independent predictors once they have been included in adjusted models, since they are likely mediators and may explain why ICU admission appears protective.
The interpretation should emphasize associations rather than causality, avoiding strong statements such as “AMR caused higher mortality.” Comparisons with prior studies would be stronger if they included direct effect size estimates rather than general statements, and reference accuracy should be checked. The strengths and limitations section should be expanded to address potential underreporting of AMR in ICD coding, the exclusion of culture-negative and polymicrobial cases, residual confounding from unmeasured variables, over-adjustment bias, and the competing risk of death when analyzing length of stay. The Discussion also repeats background information that is better suited to the Introduction; focusing on four subsections—principal findings, comparison with the literature, strengths and limitations, and implications for practice/policy—would improve clarity. Finally, the section would be strengthened by a forward-looking paragraph highlighting the need for validation of coding against microbiology data, more granular surveillance of AMR pneumonia, and the role of stewardship and infection control interventions.
Response: We are deeply grateful for this guidance. We have undertaken a major revision of this section following these recommendations closely:
- Cautious Language: We have meticulously revised the language throughout the Discussion to emphasize association rather than causation.
- Restructuring: We have reorganized the Discussion into clearer subsections: Principal Findings, Comparison with Existing Literature, Strengths and Limitations, and Implications
- Strengths and Limitations: We have significantly expanded this section to address the specific points raised. to address underreporting in ICD coding, residual confounding, over-adjustment bias, and competing risks.
- ICU Admission Interpretation: We appreciate the reviewer's comment regarding ICU admission. The reviewer suggested that ICU admission has a protective effect on mortality. The authors agree that ICU admission is an independent predictor of mortality during hospitalization. Admission to the ICU requires a rigorous assessment by the intensive care team to ensure that only patients who are candidates for intensive care interventions and therapies are admitted. Therefore, mortality rates in the ICU are lower than those on general hospital wards. Since this factor influences mortality (due to the selection of patients for ICU admission), we wanted to include it in the analysis, while acknowledging its protective effect.
- Exclusion Criteria Justification: We appreciate the opportunity to clarify this point. The strict exclusion criteria (e.g., cases with negative cultures, polymicrobial infections, or patients with a bacterial infection in addition to pneumonia) were applied to create a homogeneous group and ensure a fair comparison.
For example, including patients with additional infections would introduce significant bias, as the results could be attributed to the other infection rather than the pneumonia itself (e.g., a patient with pneumonia and a urinary tract infection versus a patient with pneumonia alone; the urinary tract infection could be the cause of sepsis, not the pneumonia, and therefore responsible for the patient's poorer prognosis, which would not be a fair comparison with a patient who only has pneumonia).
The reviewer understands that the strict exclusion criteria applied to the sample may introduce statistical bias. However, we believe that, since all these factors influence mortality, from a clinical perspective, it was preferable to exclude them to avoid skewing the comparisons in this retrospective study when comparing populations with different clinical characteristics. The authors considered this a strength of the study, and that this aspect is of such clinical importance that it justifies the significant reduction in sample size that we have accepted (Figure 4).
- Forward-Looking Paragraph: We have added a "Key recommendations and future directions" section, as suggested.
- Distinct Conclusion section
Reviewer comment: The manuscript would be strengthened by presenting a distinct Conclusion section instead of embedding it within the Discussion.
Response: We fully agree. A standalone Conclusion section has been created.
Minor Comments
- Tables and numbers
- Table 1: some counts exceed the group total (e.g., APR-DRG “Extreme” and Sepsis in AMR group). These appear to be typographical errors.
- p-values are sometimes very small (<0.0001) for minimal proportion differences. Please confirm calculations.
- Ensure consistency between text and tables (e.g., mortality IRR 1.31 vs 1.34).
- Figures
- Labels contain spelling errors (“in-hospitlal,” “lenght os stay”).
- Abbreviation inconsistency: “EVI” should be “EVCI.”
- Terminology and style
- Use “AMR” consistently.
- Italicize microorganism names.
- Replace “IC95%” with “95% CI.”
- Standardize “EU” (not “UE”) and either “US” or “U.S.” consistently.
- References
- Some citations have author name errors (e.g., “Alberti” instead of “Aliberti”).
- Reference formatting needs harmonization (punctuation, DOIs, journal abbreviations).
- Supplementary material
- Ensure appendices include complete ICD-10 code lists for reproducibility.
- Provide matching diagnostics plots if space permits.
Response: We extend our sincere thanks for the meticulous identification of typographical errors, inconsistencies, and formatting issues. We have carefully addressed each point (marked in red in the manuscript)
Appendices revised for completeness, including full ICD-10 code lists used and balance diagnostics plots where feasible. Our aim is to make this prior work, which harmonizes all the ICD-10 codes used in the study (pneumonia classification, risk factors, etc.), available to other researchers so that they can conduct reproducible studies in their respective countries (in line with the WHO request).
- Tables: corrected typographical errors, re-checked p-values, and ensured text–table consistency.
- Figures: corrected labels and abbreviations. The original figures are included in higher resolution (it seems that this was lost during the editing process)
- Terminology: ensured consistent use of “AMR,” italicized microorganism names, standardized CI reporting and regional abbreviations.
- References: corrected author name errors, harmonized formatting, and added missing DOIs.
Reviewer 2 Report
Comments and Suggestions for Authors
This is a well-structured and important study that uses a large national database from Spain (2017–2022) to assess the impact of antimicrobial resistance (AMR) on in-hospital mortality and length of stay. The dataset is robust, and the topic is highly relevant. However, several points should be addressed to strengthen the manuscript:
- The title emphasizes both mortality and length of stay, but the abstract only reports results for length of stay. Please also include the key findings on mortality.
- The discussion on AMR drivers (overuse, misuse, lack of new antibiotics) is very general. Please add more specific details, such as: Major antibiotic classes with the highest resistance rates; Key sectors contributing to AMR (human health, livestock); Known transmission routes of resistant bacteria between animals, food, and humans (One Health framework)
- The manuscript mentions the need for action but does not describe existing global frameworks.
- Define AMR, MDR, and XDR at first use.
- Italicize all bacterial names (e.g., coli, S. aureus).
- Add a short concluding paragraph summarizing key recommendations and future directions.
Author Response
Comments and Suggestions for Authors
This is a well-structured and important study that uses a large national database from Spain (2017–2022) to assess the impact of antimicrobial resistance (AMR) on in-hospital mortality and length of stay. The dataset is robust, and the topic is highly relevant. However, several points should be addressed to strengthen the manuscript:
We sincerely thank the reviewer for their thoughtful and positive assessment of our work, as well as for their constructive and insightful comments. We appreciate the time and effort dedicated to evaluating our manuscript. We have carefully considered each suggestion and believe that addressing them has significantly strengthened the paper. Our point-by-point responses to the specific comments are detailed below. All changes have been incorporated into the revised manuscript.
- The title emphasizes both mortality and length of stay, but the abstract only reports results for length of stay. Please also include the key findings on mortality.
Response: We have now included the key mortality findings in the abstract to ensure it fully reflects the study's scope and results. The added text reads: "Crude in-hospital mortality was significantly higher in the antimicrobial resistance group (18.46% vs. 10.05%, p<0.0001), with an adjusted odds ratio of 1.47 (95% confidence interval, 1.36-1.58; p<0.0001)."
- The discussion on AMR drivers (overuse, misuse, lack of new antibiotics) is very general. Please add more specific details, such as: Major antibiotic classes with the highest resistance rates; Key sectors contributing to AMR (human health, livestock); Known transmission routes of resistant bacteria between animals, food, and humans (One Health framework)
Response: We thank the reviewer for this valuable suggestion, which enhances the article. To address this point comprehensively, we have added a new paragraph to both the introduction and the discussion sections, incorporating the requested details.
These new sections describe the main bacterial AMR pathogens of global concern, based on the 2024 WHO Bacterial Priority Pathogens List. The discussion also addresses the "One Health" concept, as suggested.
- The manuscript mentions the need for action but does not describe existing global frameworks.
Response: This is an excellent point. We have added a new we agree that describing existing frameworks strengthens our call to action.
Following the reviewer's suggestion, we have added a new paragraph in the Discussion section that briefly outlines relevant global initiatives, including The WHO Global action plan on antimicrobial resistance", "A European One Health Action Plan against Antimicrobial Resistance" (UE), "National Action Plan for Combating Antibiotic-Resistant Bacteria" (U.S.) and the "One Health" approach promoted by quadripartite organizations (WHO, FAO, WOAH, UNEP).
- Define AMR, MDR, and XDR at first use.
Response: We thank the reviewer for this suggestion. We now define MDR (Multidrug-Resistant), XDR (Extensively Drug-Resistant), PDR (Pan Drug-Resistant) and Difficult-to-Treat Resistance (DTR) at their first mention in the introduction.
- Italicize all bacterial names (e.g., coli, S. aureus).
Response: We appreciate the reviewer's attention to detail on this matter. We have carefully reviewed the entire manuscript and corrected the formatting by italicizing all scientific names of microorganisms.
- Add a short concluding paragraph summarizing key recommendations and future directions.
Response: We followed this recommendation and added a dedicated section entitled key recommendations and future directions. This new section succinctly summarizes the key recommendations arising from our findings and suggests valuable directions for future research.
Reviewer 3 Report
Comments and Suggestions for Authors
The paper entitled “Impact of antimicrobial-resistant bacterial pneumonia on in-hospital mortality and length of stay: a nationwide observational study in Spain based on the International Classification of Diseases, version 10” analyzes the impact of pneumonia caused by antimicrobial-resistant bacteria on hospital mortality and length of hospitalization in Spain, using a national database based on ICD-10 coding.
This manuscript makes a useful and valuable contribution to the knowledge and understanding of the subject matter. Authors clearly showed that their research is replicable, and methodology was described in enough detail. The authors followed the principles of good practice and the results obtained are clearly explained, the conclusions follow a clear logic and are supported by adequate references.
- Please look at the title, is version 10 really necessary or is it an oversight?
- I recommend that the authors create a separate section called "Limitations of the study". Although some limitations are already mentioned in the Discussion, presenting them in a structured paragraph would improve transparency and strengthen the validity of the manuscript. Specifically, the authors could emphasize:
Potential misclassification due to ICD-10 coding - Because identification of cases and controls relied entirely on administrative codes, coding errors or underreporting could have led to patient misclassification, despite the overall robustness of the RAE-CMBD database.
Lack of data on antimicrobial treatment - The database does not provide information on the appropriateness, timing, or duration of empiric antibiotic therapy, which are key determinants of patient outcomes and may have influenced mortality estimates.
Heterogeneity in microbiological confirmation - Identification of AMR was based on routine clinical practice at multiple centers, without standardized sampling or laboratory confirmation by the study team. This heterogeneity could introduce variability and bias in the classification of resistant cases.
Author Response
Comments and Suggestions for Authors
The paper entitled “Impact of antimicrobial-resistant bacterial pneumonia on in-hospital mortality and length of stay: a nationwide observational study in Spain based on the International Classification of Diseases, version 10” analyzes the impact of pneumonia caused by antimicrobial-resistant bacteria on hospital mortality and length of hospitalization in Spain, using a national database based on ICD-10 coding.
This manuscript makes a useful and valuable contribution to the knowledge and understanding of the subject matter. Authors clearly showed that their research is replicable, and methodology was described in enough detail. The authors followed the principles of good practice and the results obtained are clearly explained, the conclusions follow a clear logic and are supported by adequate references.
We are sincerely grateful to Reviewer 3 for their highly positive and encouraging assessment of our manuscript. The observation that it "makes a useful and valuable contribution" is truly motivating for our team. We also thank the reviewer for their constructive suggestions, which have been instrumental in enhancing the quality and clarity of our work. Our point-by point responses are detailed below,
We believe the revisions have significantly strengthened the manuscript.
- Please look at the title, is version 10 really necessary or is it an oversight?
Response: We thank the reviewer for this astute observation. Upon review, we agree that specifying "version 10" is redundant in this context. The updated title is now: "Impact of antimicrobial-resistant bacterial pneumonia on in-hospital mortality and length of stay: a nationwide observational study in Spain"
- I recommend that the authors create a separate section called "Limitations of the study". Although some limitations are already mentioned in the Discussion, presenting them in a structured paragraph would improve transparency and strengthen the validity of the manuscript. Specifically, the authors could emphasize:
Potential misclassification due to ICD-10 coding - Because identification of cases and controls relied entirely on administrative codes, coding errors or underreporting could have led to patient misclassification, despite the overall robustness of the RAE-CMBD database.
Lack of data on antimicrobial treatment - The database does not provide information on the appropriateness, timing, or duration of empiric antibiotic therapy, which are key determinants of patient outcomes and may have influenced mortality estimates.
Heterogeneity in microbiological confirmation - Identification of AMR was based on routine clinical practice at multiple centers, without standardized sampling or laboratory confirmation by the study team. This heterogeneity could introduce variability and bias in the classification of resistant cases.
Response: We thank the reviewer for this excellent recommendation, which we believe substantially enhances the manuscript's rigor and transparency. We fully agree that a dedicated section provides a clearer and more structured discussion of the study's limitations.
Following this advice, we created a new subsection titled "Limitations" within the Discussion. In this section, we have systematically addressed each of the suggested issues:
- We have expanded the discussion on potential misclassification due to ICD-10 coding.
- We have explicitly acknowledged the lack of data on antimicrobial treatment and its potential impact on our outcomes.
- We have detailed the implications of heterogeneity in microbiological confirmation across participating centers.
We are confident that this new, focused section provides a more comprehensive and transparent appraisal of the study's constraints, thereby strengthening the validity of our findings and conclusions
Round 2
Reviewer 1 Report
Comments and Suggestions for Authors
I have carefully reviewed the revised manuscript by Oterino-Moreira et al. and compared it with my earlier comments. The authors have made substantial improvements in clarity, transparency, and framing, and I appreciate their constructive and detailed responses. The manuscript is significantly improved.
That said, several important issues remain that must be addressed before the manuscript is suitable for publication.
Study design terminology (retrospective cohort vs. case-control)
The Abstract and Methods now correctly describe the study as a retrospective cohort. However, the title still uses the generic term “nationwide observational study,” and several sections, including Figure 4, continue to present the groups as “cases” and “controls.” This inconsistency may confuse readers about the underlying study design. Recommendation: Update the title to explicitly reflect “retrospective cohort study” and remove all “case/control” terminology throughout the manuscript, including figure legends and appendices.
Outcome modelling (mortality analysis)
The manuscript presents logistic regression with odds ratios as the primary method for mortality, while modified Poisson regression appears only as a sensitivity analysis. The response letter, however, suggested Poisson regression was the main approach. This discrepancy requires clarification, as it may mislead readers about the primary analytic framework. Recommendation: Clearly state in the Methods whether logistic regression or modified Poisson regression is the primary analysis. If logistic regression is retained as primary, present Poisson regression results explicitly as a sensitivity analysis.
Interpretation of associations
The revised Discussion is more structured, but causal language remains, with statements such as “AMR leads to worse outcomes” and references to “independent risk factors.” Such phrasing risks overstating the strength of inference. In addition, ICU admission continues to appear protective in adjusted models, which most likely reflects mediation or selection bias rather than a true protective effect.
The new material describing the WHO Global Action Plan, the EU and U.S. national strategies, and the One Health Joint Plan provides relevant background on international AMR policy frameworks. However, in their current form, these paragraphs are lengthy, read more like policy summaries, and feel somewhat disconnected from the study’s findings. Much of this information belongs in the Introduction rather than in the Discussion, where it distracts from the manuscript’s central contribution. In particular, the detailed enumeration of the six One Health action tracks is excessive and not clearly linked to the present analysis of AMR pneumonia in Spain.
Conclusion section
A distinct Conclusion section has been added, which is appreciated. However, in its current form it largely repeats material already presented in the Discussion and reads somewhat generic. It does not provide a strong synthesis or forward-looking perspective.
Also, several smaller points require attention to improve accuracy and consistency. In Table 2, the APR-DRG “Extreme” category lists 21,815 AMR cases despite the AMR cohort being only 6,017 patients; this appears to be a typographical error (likely 1,815) and should be corrected. In terms of style, “IC95%” should be replaced with “95% CI,” “Sensibility analysis” with “Sensitivity analysis,” “Standar error” with “Standard error,” and “Quintil” with “Quintile.” Terminology should be standardized, using “EU” rather than “UE,” and “US” or “U.S.” consistently. Likewise, the manuscript should use “AMR” uniformly (Figure 4 currently refers to “antibacterial resistance”). All Latin microorganism names should be italicized.
Best regards,
